# Multi-Risk Assessment of Mine Lithium Battery Fire Based on Quantitative Factor Characterization

**DOI:** 10.3390/ijerph20010456

**Published:** 2022-12-27

**Authors:** Kuikui Li, Yanming Wang, Yongchang Zhang, Shasha Wang, Xiangyu Zou

**Affiliations:** 1School of Safety Engineering, China University of Mining & Technology, Xuzhou 221116, China; 2Research Center of Intelligent Ventilation, China University of Mining & Technology, Xuzhou 221116, China

**Keywords:** risk assessment, mine fire, lithium battery, quantitative characterization

## Abstract

As a large number of new energy is employed as the driving force for the operation and transportation machinery of underground space projects, the lithium battery load in confined spaces, such as working faces, roadways and tunnels increases in geometric progression, and the coupled risks of heat damage and smoke poisoning caused by possible fires become more serious. In this paper, experimental and numerical methods were implemented to study the propagation mechanism of heat- and mass-induced disasters under catastrophic conditions, and a quantitative characterization model of multiple risk factors of thermal runaway and toxic gas diffusion of battery fire was proposed. The fuzzy analytical hierarchy process (FAHP) was conducted to calculate and grade the risk of lithium battery fire in a typical mine working face under multiple factors, including hazard source, personnel, working environment and emergency response. In addition, a quantitative early warning and control model was established for identified high-risk probability events. The results promote the quantitative and scientific development of multiple risk assessment and decision-making of confined space fire.

## 1. Introduction

On 12 May 2020, China officially released three mandatory national standards for electric vehicles, including the Standard of Safety Requirements for Power Batteries for Electric Vehicles, which added the relevant requirements for thermal diffusion of battery systems. It read that ‘5 min before the danger in the passenger compartment caused by thermal diffusion of a battery pack or system due to thermal runaway of a battery, a heating event alarm signal should be provided (alarm for the whole vehicle to remind passengers to evacuate)’. Hence, the implementation of the new national standard put forward more requirements for battery thermal runaway and thermal diffusion management.

With the improvement of coal mining automation and the promotion of new energy technologies, lithium batteries are increasingly used in underground power and transportation systems, especially in large mining enterprises with a capacity of 10 million tons in northwest China. Under the complex underground operation and environmental conditions, the hidden danger of high-load lithium battery has brought new challenges to the prevention and control of external fire in coal mines.

For the thermal safety of lithium battery itself, as a research hotspot, extensive research has been carried out on thermal runaway propagation of module battery, battery material stability and other aspects, and it is recognized that thermal runaway of lithium battery is usually caused by different abuse conditions, including a. mechanical factors such as acupuncture, extrusion, collision, b. thermal factors such as external high temperature, flame burning and c. electrical factors such as overcharge and discharge, etc. [1,2,3]. The above conditions may lead to the fusing of the lithium battery diaphragm and a short circuit between the positive and negative electrode materials. When the electrode materials have a variety of exothermic reactions under the action of high temperature, the continuous accumulation of heat would cause a battery fire [4]. In the field of lithium battery fire risk assessment, cone calorimeter has been used to measure the heat release rate, total smoke generation, total heat release and other parameters of battery diaphragm, positive and negative electrodes under full charge state. Based on the data set composed of heat and toxicity, the comprehensive assessment index system of lithium battery fire risk has been established by using analytic hierarchy process (AHP) [5]. Using the fault tree analysis method (FTAM), based on the fire triangle model, the basic events of lithium battery fire accidents and their importance has been analyzed [6]. Meanwhile, considering the characteristic parameters and reaction mechanism, the risk index of thermal runaway of lithium battery has been proposed based on the risk assessment theory and applied to the quantitative safety assessment of lithium battery [7,8]. Furthermore, the semi quantitative method (LEC) has been employed to conduct safety assessment on the fire caused by the lithium battery disposal process, classify the hazard level of the consequences of fire accident and propose the main prevention directions of lithium battery safety accidents [9,10].

It should be pointed out that the above research mainly focused on the safety risks caused by the thermal runaway of lithium battery itself, and there are relatively few studies on the hazard of lithium battery fire to life in complex environments. In fact, burning substances in industrial or public scenes, as is the cause of disasters, only generates the sources of radiant heat and toxic gas, while the risk factors threatening personal safety are more directly related to heat and mass transport conditions, diffusion environment, safety technology and management measures. Therefore, it is an unsolved practical problem to objectively and accurately evaluate the safety risk of lithium battery fire in mine production process by considering various factors. Lithium battery fire is a new form of external fire in coal mine. At present, the main means of mine fire risk assessment is to build an incomplete quantitative assessment model based on small sample statistics theory [11]. According to the comprehensive theory of accident causes, combined with the actual characteristics of the coal mine environment, the existing works have analyzed the risk factors that affect the coal mine external fire, constructed the coal mine external fire risk index system and evaluation model, divided the risk level and discussed its governance scheme [12]. For example, it is proposed to establish optimization evaluation model with matter-element and extension theory, and present the risk assessment model of mine fire by using set-valued statistics, fuzzy comprehensive evaluation (FCE), or combining fuzzy analytic hierarchy process (FAHP) and artificial intelligence (AI) algorithms [13,14,15,16,17]. However, these evaluation models have not been applied to the fire safety of lithium battery in coal mine.

Nowadays, the electrical load of mining equipment is increasing day by day, and the risk of lithium battery thermal runaway is on the rise significantly. As the power or transport equipment operates mostly in the main ventilation roadway or working face, sudden battery fire is highly random, and a large amount of toxic gas spreads rapidly with ventilation (up to 5–10 m/s), and the fire smoke invasion area further expands in the complex roadway system, which will lead to explosion and cause serious casualties and property losses [18]. Therefore, the scientific and accurate evaluation of the safety risk of lithium battery under dynamic operation conditions needs to first solve the quantitative problem of multiple hazard sources (thermal runaway, toxic gas) and their diffusion risk, as well as introduce the mapping relationship between disaster characteristics and emergency escape in complex environments, and comprehensively consider various factors such as human-machine-environment-management, so as to establish a reliable and digital fire risk evaluation and safety management model of mining lithium battery.

In this study, a multiple risk assessment model for the possible occurrence of lithium battery fire disaster in the confined space of the mine is established, considering the potential coupling hazards of thermal runaway injury and toxic smoke poisoning. An innovative analysis method which combines the quantitative factor characterization of hazard sources and FAHP is proposed to reduce the subjective deviation of weight determination.

## 2. Materials and Methods

### 2.1. Risk Management Process

Risk management is a systematic cycle process, mainly including risk identification, risk assessment and risk response. The purpose of risk management is to take timely measures to avoid risks or reduce the probability of risk occurrence as much as possible when risk events have not yet occurred; at the time of or after the risk event, minimize the loss caused by the risk event or within the acceptable range, and maximize the security benefits as much as possible [19].

(1)Risk identification

Risk identification is the process of comprehensively and carefully searching for various uncertain factors that may exist in the internal and external environment item by item. It is the process of checking the internal and external environment of the research object, identifying various factors that may have positive or negative effects on the research object, and defining and scientifically classifying various factors. Risk identification is the basis and premise of risk management and has a direct impact on the final effect of risk management. As the field of risk management research continues to deepen, there are various methods for risk identification, both subjective and objective, and the more widely used risk identification methods are literature analysis, brainstorming, report analysis and expert interview.

(2)Risk assessment [20]

Risk assessment is the core part of risk management theory and plays a key role in the process of risk management. Risk assessment is the process of establishing a standard system of risk assessment based on the results of risk identification, judging whether these risk factors will affect the realization of objectives, and evaluating the degree of impact. A dynamic and continuous process of risk assessment is conducive to the normalization of risk management, thus providing a basis for taking countermeasures in advance and making correct risk management decisions. Risk assessment methods include qualitative analysis, quantitative analysis and comprehensive analysis by combining the two. However, the combination of qualitative and quantitative methods is commonly used. Common risk assessment methods are summarized as Table 1.

(3)Risk response

Risk response is the process of taking certain measures to effectively avoid or reduce losses caused by risks based on the results of risk identification and evaluation. Risk response is a key step in risk management, which runs through the whole process of enterprise production and operation. The quality of risk control is directly reflected in the quality of the enterprise’s comprehensive benefits.

### 2.2. Identification Method of Fire Risk

In order to accurately identify hazard indicators for mine lithium battery fire, it was necessary to select the most appropriate risk assessment method based on the characteristics and purposes of the research object. Here, an analysis method of quantifying qualitative indicators called fuzzy analytic hierarchy process (FAHP) [21] was proposed. The method uses the mathematical theorem of the fuzzy matrix to make decisions on people, things, things, plans and other multi factor indicators. Based on the objective facts, it conducts quantitative analysis on things that do not have laws, and finally obtains the comprehensive evaluation results. It is a method used to convert the traditional qualitative objects into quantitative representations. This method is used to study the problem of uncertainty. The FAHP improves the original single analytic hierarchy process and improves the accuracy and integrity of decision-making [22,23].

FAHP is favored for its characteristics of combining qualitative and quantitative methods to deal with various evaluation factors, as well as its advantages of being systematic, flexible and concise. In addition, FAHP is widely used in all walks of life due to its theoretical integrity, structural rigor, simplicity in solving problems and obvious advantages in solving unstructured decision-making problems, especially in the field of complex social sciences.

### 2.3. Quantification of Risk Indicators

Although FAHP has obvious advantages, this method still requires subjective judgment from experts, the scoring matrix may have logical errors or omissions, and there would be deviations from human thinking when checking the consistency of machine operations. At this time, A quantitative model presented in Section 3 was introduced to calculate partial risk factors in the FAHP model and optimize the indexes weight calculation, so as to reduce errors [24]. In order to obtain quantitative risk indicators scientifically and objectively, this paper proposed to establish risk failure probability data by means of lithium battery thermal runaway test and mine external cause fire simulation.

#### 2.3.1. Experiment Setup

The risk of thermal runaway of mining lithium battery was studied by experimental methods [25,26,27]. The lithium iron phosphate battery used in the mine transport vehicle was produced by China Innovation Aviation Corporation, with LiFePO_4_/graphite as the electrode. The LIB had a nominal capacity of 68 Ah and a nominal voltage of 3.22 V. Lithium batteries with expected SOC of 100%, 75% and 50% were prepared by constant current/constant voltage method; the expected number of lithium batteries were prepared by the wire binding method, and the number of lithium batteries were 1, 5, 9 and 13, respectively; the lithium battery with the expected distribution interval were prepared by vertically arranging two batteries, and the distribution intervals were 0.02 m, 0.05 m and 0.08 m, respectively. The schematic diagram of the experimental device is shown in Figure 1.

#### 2.3.2. Simulation Model

Under the ventilation condition of the working face, the danger of toxic gas diffusion caused by mining lithium battery fire was simulated and analyzed by the numerical method of flow and heat transfer. FDS simulation was involved in the study on the migration of fire smoke of lithium battery under multiple working conditions [28,29,30]. The code can simulate the solution and obtain a series of data such as temperature, CO concentration, CO_2_ concentration, O_2_ concentration and visibility at relevant measuring points. It effectively solves the large eddy simulation of the development of fire dynamics. Users can adjust the accuracy and speed of the calculation by setting the grid density. Through FDS simulating, the fire scene could be accurately simulated, and the real fire scene could be even better simulated.

In this paper, a fire scene was set up in the underground roadway of the mine working face. The road cross section was 4 m × 4 m, the measuring point was 100 m away from the fire source. The physical model of road fire is shown in Figure 2. By simplifying the lithium battery fire, according to a large number of experimental studies on GV, the content and composition of smoke generated by lithium battery fire were similar to plastic (PVC) fire. Therefore, the fire source scale was set as 10 MW, the type was t^2^ fast fire defined in ISO/TS 16733, the fire growth coefficient was 0.04689 kW/s^2^ and the PVC reaction in the FDS instructions was used to set CO generation.

## 3. Risk Assessment Model

### 3.1. Risk Factor Contribution Calculation

Based on fire and smoke risk assessment, the FED model—which is an asphyxiant gas model—could judge the hazard of toxic smoke to personnel according to the relationship between the concentration of CO and disabling time. It was obtained according to the IS013571 [31] literature:


(1)
$$FED=\underset{4_{1}}{\sum^{t_{2}}}\;\frac{\left[CO\right]}{35000}\;\;\;\Delta \;\;\;t+\underset{t_{1}}{\sum^{t_{2}}}\;\frac{\exp⟮\frac{HCN}{43}⟯-1}{220}\;\;\;\Delta \;\;\;t$$


Equation (1) gives the product of the concentration of *CO* and *HCN* gas that makes people disabled and the time they take. In mine external fire, *HCN* damage is generally not considered, so the above formula can be simplified as:


(2)
$$FED=\underset{t_{1}}{\sum^{t_{2}}}\;\frac{\left[CO\right]}{35000}\;\;\;\Delta \;\;\;t$$


Where, the compromising tenability dose, (*C·t*), for *CO* of 35,000 uL/min was obtained from experiments on juvenile baboons subjected to an escape paradigm. The equation means that a dose of 35000 ul/min would produce approximately 30% blood carboxyhaemoglobin saturation in humans having average adult body weight and a respiratory minute volume of 20 L/min [32].

This model conducted risk analysis from the perspective of the impact of accidents. By setting the FED threshold, the time required for CO under different working conditions from the occurrence of disasters to the threshold could be quantified by experiment or simulation. Therefore, the contribution rate *K* was defined as the difference ratio of the time to reach the threshold t and the difference ratio of the impact factor *F*.
(3)$$K=\frac{\Delta t}{\Delta F}$$

The above formula was used to calculate the contribution value of the unit factor of the same factor under different working conditions, and finally to take the average value of the whole working condition. Then the factor contribution value was used to calculate the factor weight distribution by the normalization method. The relative importance score of two factors was obtained through the ratio of contribution rate value, and the judgment matrix was constructed.

### 3.2. FAHP Model

The core of FAHP is to construct the fuzzy consistency judgment matrix [33,34,35,36,37,38]. Compared with the AHP method, it is more consistent with the decision-making goal, and the realization process is more convenient and faster. The specific process is shown in Figure 3.

The application steps of FAHP was divided into two parts: hierarchical analysis and fuzzy comprehensive evaluation. The details are as follows:(1)Analyzed the relationship between the risk factors, divided the risk factors into different levels, and finally built the risk evaluation index system;(2)The importance of each index at the same level with respect to a criterion at the previous level were compared in pairs, and the pairwise comparison judgment matrix was constructed;(3)According to the judgment matrix, calculated the relative weight of each index to the index of the first level;(4)Ranked the risk weights and conducted a consistency test to eliminate the impact of logical errors on the evaluation results;(5)Determined the impact path and degree of each risk factor on the evaluation results;(6)Invited a certain number of experts to judge the impact of various risk factors on the evaluation results. Then, the evaluation judgment matrix was constructed;(7)The final evaluation value was obtained by comprehensive evaluation judgment matrix and weight set and fuzzy comprehensive calculation.

#### 3.2.1. Calculation Matrixes

(1)Definition of fuzzy matrix

For a matrix $B=\left(b_{ij}\right)m\times n$, if $0\leq b_{ij}\leq 1(i=1,2,\ldots ,m;j=1,2,\ldots ,n)$, it is called a fuzzy matrix.

(2)Definition of Fuzzy Complementary Judgment Matrix

Obviously, each factor is qualitative. If a certain method is not used for pretreatment, mathematical tools cannot be used for calculation. In this way, the solution of the model cannot be completed, and the relationship between various factors is difficult to study. Therefore, it was necessary to adopt quantitative analysis method to preprocess the influencing factors, mainly to conduct fuzzy analysis on the importance of each factor, and then the fuzzy judgment matrix $A=\left(a_{ij}\right)m\times n$ was obtained. If the obtained matrix met the following requirements, it was called the fuzzy complementary judgment matrix. requirement:(4)$$\begin{array}{c} a_{ii}=0.5(i=1,2,\ldots ,n)\\ a_{ii}+a_{ij}=1(i=1,2,\ldots ,n) \end{array}$$

(3)Establish weight judgment matrix

Therefore, it was necessary to assign weights to each element manually for the first time. Because it was an artificial subjective allocation, errors could not be avoided. However, we tried to listen to expert advice and searched for enough information before allocating the weight of elements. According to the comparison scale table in Table 2, the following fuzzy judgment matrix was obtained.

Requirement:(5)$$A=\left[\begin{array}{cccc} a_{11} & a_{12} & \ldots & a_{1n}\\ a_{21} & a_{22} & \ldots & a_{2n}\\ \vdots & \vdots & \ddots & \vdots \\ a_{n1} & a_{n2} & \ldots & a_{nn} \end{array}\right]$$

Through the establishment of the above judgment matrix, the relationship between the indicators was obtained according to the 1–5 scale method (Table 2), and it was easy to find that the matrix had the following mathematical characteristics:(6)$$\left\{\begin{array}{l} a_{ij}>0\\ a_{ij}=\frac{1}{a_{ji}}\\ a_{ij}=1 \end{array}\right.$$

#### 3.2.2. Weight Calculation

After the feature vector W was calculated, it could not be used directly, because it would have produced large errors and violated the scientific principle of model establishment, so it also needed to be preprocessed to make it meet $\sum_{ti}^{n}\;W_{i}=1$, so that the weight data of each index could be accurately obtained. There are many methods for finding eigenvectors, such as geometric average method, arithmetic average method, eigenvalue method, etc. Take geometric average method to calculate eigenvector *W* as an example:

(1)Step 1: Calculate the product of each line of the judgment matrix ${\left(M_{1},M_{2},\ldots ,M_{i},\ldots ,M_{n}\right)}^{T}$;(2)Step 2: calculate the value *Y* of the *n*-th root of *M_i_*;(3)Step 3: Normalize the root of the *n*-th power, $W_{i}=Y_{i}/\sum_{i=1}^{n}\;Y_{i},W={\left(W_{1},W_{2},W_{3},\ldots ,W_{n}\right)}^{T}$ is the obtained eigenvector, that is, the weight of the element;(4)Step 4: Calculate the maximum eigenvalue: λ *_max_*.

#### 3.2.3. Consistency Inspection

The selection of evaluation indicators was preliminarily selected through expert consultation, network search and other methods, so it had a great human orientation. This error could not be eliminated, and could only be reduced continuously, which affected the calculation of the following eigenvalue. Therefore, after c is calculated, a consistency test was required to reduce the error and improve the accuracy of the results. The specific inspection methods were as follows.

Calculation consistency CI:(7)$$CI=\left(\lambda_{\max}-n\right)/(n-1)$$

Calculate the consistency ratio CR:(8)CR=CI/RI

Among them, RI is the critical value of the average random consistency index test, and the random consistency index RI value given by Professor Saaty is as Table 3.

When CR < 0.1 is within the range specified in the academic field, it can be determined that the construction of the matrix is consistent with the principle and is used in the actual research. Of course, the smaller the CR is, the higher the consistency between the judgment matrix and the decision goal is.

### 3.3. FCE Medol

As we all know, in real life, there are almost no problems that are only affected by a single factor. Generally, they are affected by many factors. Therefore, it is necessary to design a new mathematical model to solve the multifactor problem. The multi-level fuzzy evaluation method was introduced here [39,40,41]. The specific methods were as follows:

(1) Determine the factor set of the evaluation object.
(9)$$U=\left\{U_{1},U_{2},U_{3},\ldots ,U_{n}\right\}$$

(2) Division of factor set.

Dividing factor set $U=\left\{U_{1},U_{2},U_{3},\ldots ,U_{n}\right\}$ into subsets; $U={\cup U}_{i},U_{1}=\left\{u_{1},u_{2},u_{3},\ldots ,u_{k}\right\}$ is the first level factor set, and so on.

(3) Determine the evaluation set of factor set *U*.
(10)$$V=\left\{V_{1},V_{2},V_{3},\ldots ,V_{m}\right\}$$

(4) Assuming that.

$U_{i}=\left\{u_{1}^{i},u_{2}^{i},u_{3}^{i},\ldots ,u_{n_{i}}^{i}\right\}\;\left(i=1,2,3,\ldots ,k;n_{1}+n_{2}+n_{3}+\ldots +n_{k}=n\right)$ is a two-layer factor set, the above factor evaluation method can be used for each factor in the set to obtain a single factor evaluation matrix as follows:(11)$$R_{i}=\left[\begin{array}{cccc} r_{11}^{i} & r_{12}^{i} & \ldots & r_{1n}^{i}\\ r_{21}^{i} & r_{22}^{i} & \ldots & r_{2n}^{i}\\ \vdots & \vdots & \ddots & \vdots \\ r^{i} & r^{i} & \ldots & r_{mn}^{ii} \end{array}\right]$$

Assuming that the weight of $\left\{R_{i}=\left\{r_{1}^{i},r_{2}^{i},r_{3}^{i},\ldots ,r_{n_{i}}^{i}\right\}\right.$ is $W_{i}=\left\{w_{1}^{j},w_{2}^{j},w_{3}^{j},\ldots ,w_{n_{i}}^{j}\right\}$, we can get the following evaluation results:(12)$$B_{i}=W_{i}\circ R_{i}=(i=1,2,\ldots ,k)$$

(5) Then the set $U=\left\{u_{1},u_{2},u_{3},\ldots ,u_{k}\right\}$ makes a comprehensive evaluation. Let the weight vector of $U=\left\{u_{1},u_{2},u_{3},\ldots ,u_{k}\right\}$ be $W=\left(w_{1},w_{2},w_{3},\ldots ,w_{k}\right)$, and the following matrix is obtained:(13)$$B={\left(b_{1},b_{2},b_{3},\ldots ,b_{k}\right)}^{T}$$

(6) Finally, the comprehensive evaluation criteria are:(14)$$B_{1\times m}=W_{1\times m}\circ R_{k\times m}$$

## 4. Results and Discussion

### 4.1. Indicators Identification of Mine Battery Fire

This study focused on the safety problem of lithium battery fire in the mine working face. In the case study, taking a large state-owned coal mine as a typical model. The mine is located in Shaanxi province in northwest China with a mining area of 5.489 square kilometers and an annual output of 10 million tons. 

The danger level of gas outburst in this mine is not high, and the burial depth is shallow. The opening way of mine is “two inclined shafts and one vertical shaft”, which the inclined shaft undertakes the auxiliary transportation task and personnel passage, and the main shaft is responsible for ventilation and centralized coal production. The lifting and transportation equipment adopts advanced and reliable large elevators, rail cars, monorail cranes, long distance belt conveyors, gear cars and other vehicles for auxiliary transportation. Long-wall fully mechanized mining is mainly used for coal mining in the working face, with high mechanization, many power-consumption items, heat hazard factors such as surrounding rock heat dissipation and mechanical heating and other potential safety hazards. 

With reference to the risk identification methods in 2.1 and the literature on the study of exogenous fires in coal mines using the FAHP method, and taking into account the characteristics of lithium battery fires, literature analysis was conducted to identify the risk indicators for lithium battery fires in mines, and the identified risk factors were classified and then the opinions and suggestions of relevant experts and scholars were sought in the form of interviews to screen out the unreasonable factors. The primary indicators were classified from the four aspects of human-machine environmental management for the risk of lithium battery fires in coal mines, and the secondary indicators were obtained by screening according to the correlation and mutual independence criteria with the primary indicators.

For the case study, various risk factors for lithium battery fires in the mine have been analyzed based on the power feedback from workers, underground vehicle conditions, mine heat damage assessment and roadway layout and management, as shown in Table 4.

Based on the index stratification in Table 4, it was easy to build an analytic hierarchy process model that affects the fire risk of lithium batteries. Among them, the first level index was the criterion level, and the second level index was the decision-making level. The comprehensive factor A was all the factors that affect the fire risk of lithium battery.

a. Target layer: lithium battery fire risk factor (A).

b. Baseline layer: the direct source of risk, including personnel factor (B1), hazard factor (B2), crisis management factor (B3) and environmental factor (B4).

c. Utility layer: corresponding to the specific source of the specific risk factors generated by the benchmark layer, these factors are uniformly numbered and represented by symbols B11, B12, B13, B22, B23, B31, B32, B33, B34, B41, B42, B43 respectively.

The structure topology relationship between each level is shown in Figure 4.

### 4.2. Quantitative Study of Risk Factors

#### 4.2.1. Risk Source Factors of Lithium Battery Fire

In order to better reproduce the process of thermal runaway of lithium battery, the fixed flux source was used to heat the lithium battery, so that thermal runaway of lithium battery occurred. Figure 5 shows the physical picture of lithium battery thermal runaway ignition and instrument acquisition interface. 

When the lithium battery pack was heated to the burning state and the smoke was released, the temperature, composition and concentration of smoke at the fire source and at the simulated tunnel pipeline location were observed online, and the risk value of fire smoke concentration and its change rule with time were obtained. For the risk source factors characterization, the experiment was conducted with lithium iron phosphate vehicle batteries in the following three aspects: distribution distance, quantity and state of charge. Although, the influencing factors of lithium battery types were ignored. The Equation (2) of FED model was employed to calculate the time when they reach the threshold as seen in Figure 6.

#### 4.2.2. Working Environmental Factors

FDS software was used to simulate the situation of lithium battery fire in the roadway. Figure 7 was the cloud diagram of simulation results.

In the work environmental characterization, the following three aspects were analyzed for the external fire in mine working face, namely, secondary hazards, ventilation conditions and geographical conditions. The secondary hazards mainly caused the combustion of tires and surrounding coal, which also produced a large amount of CO, but there was a slight time lag. This study considered that it occurred at the same time, hence, it was assumed that the generation of secondary hazard was equivalent to the increase in fire source scale; ventilation conditions were measured by roadway wind speed only when local ventilation and air leakage were ignored; the geographical conditions ignored the characteristics of surrounding rock, and the roadway dip angle was used to measure. Equation (2) was used to calculate the time when they reached the threshold as seen in Figure 8.

### 4.3. Quantitative Characterization of Indicators Weight

#### 4.3.1. Primary Indicators

In order to assess the fire risk of lithium battery in the mine, we developed a matching questionnaire, and divided the indicator scoring interval into five grades: “very poor, poor, average, good, and very good”, with the corresponding scores being 1, 2, 3, 4, and 5 in turn. Work safety experts from enterprises, universities, safety supervision departments and other institutions were selected as the survey objects. Experts had been engaged in work safety for more than 15 years on average, and they gave answers based on their own work background and the actual situation of the enterprise’s work safety. The score of the corresponding scoring range was in direct proportion to the influence degree of the element level. The higher the score, the greater the influence on the indicator. A total of 10 expert questionnaires were issued and 10 valid questionnaires were recovered. See Appendix A for the final parameter values of indicators calculated according to the questionnaire. 

Because different evaluation indicators had different effects on the final evaluation results, this paper determined the weight of the set indicators of the evaluation indicator system by comprehensively using expert scoring method and analytic hierarchy process. The specific steps and results were as follows.

A total of 10 experts gave a score, using a scale of 1–5 points. For example, factor A is very important to factor B. At this point, 5 points will be given, and factor B will be 1/5 of factor A, that is, 0.2 points. Factor A is more important than factor B, so 3 points will be given; The importance of factor A is the same as that of factor B, which is 1 point. A total of 10 experts gave scores.

When calculating the weight with the analytic hierarchy process, take the weight calculation of the secondary immunity index of the antibody index as an example. As shown in Table 5, a judgment matrix needed to be constructed for B1–B4 first, and the judgment matrix was obtained by comparing two elements. The construction method of judgment matrix was as follows: calculate the average value of each analysis item, and then divide the average value to obtain the judgment matrix. The greater the average value was, the higher the weight would be. AHP is usually used for experts to score the importance of indicators.

Secondly, the characteristic root and weight (Table 6) were calculated. The eigenvector value, weight value and maximum eigenvector were calculated; then the maximum characteristic root value CI, which was used for the consistency check of the maximum characteristic in the next step, was obtained.

The consistency test shows that CR = CI/RI = 0.003 ≤ 0.1, so the judgment matrix had satisfactory consistency. The final weight vector was (0.144, 0.196, 0.258, 0.402), and the weight distribution of primary indicators is shown in Figure 9.

#### 4.3.2. Secondary Indicators

The judgment matrix construction of the weight of the indicators in the second layer was divided into two categories. First, the personnel factor B1 and the crisis management factor B3, which were difficult to quantify, were still used to construct the judgment matrix in a fuzzy way using the above method and expert scoring. The other aspect was that, for hazard source factor B2 and environmental factor B4, instead of using the scoring value of experts, the contribution value method was constructed to calculate the unit contribution of the unit influencing factor to the accident consequence based on the FED asphyxiation model from the perspective of the accident consequence using the digital method, and the evaluation matrix was constructed scientifically and visually.

The contribution rates of each factor index according to the FED model and Equation (3) were calculated, and the weight distribution of the contribution rate was normalized, as shown in Table 7.

The judgment matrix of the secondary indicators under the primary indicators B1, B2, B3 and B4, was constructed and the secondary indicator matrix was obtained, as shown in Table 8.

Normalized weight of all indicators can be obtained:$$\left\{\begin{array}{c} \beta 1={\left(\begin{array}{ccc} B11 & B12 & B13 \end{array}\right)}^{T}={\left(0.246\;0.477\;0.277\right)}^{T}\\ \beta 2={\left(B21\;B22\;B23\right)}^{T}={\left(\begin{array}{ccc} 0.150 & 0.603 & 0.247 \end{array}\right)}^{T}\\ \beta 3={\left(\begin{array}{cccc} B31 & B32 & B33 & B34 \end{array}\right)}^{T}\\ ={\left(\begin{array}{cccc} 0.371 & 0.213 & 0.180 & 0.236 \end{array}\right)}^{T}\\ \beta 4={\left(B41\;B42\;B43\right)}^{T}={\left(\begin{array}{ccc} 0.544 & 0.265 & 0.191 \end{array}\right)}^{T} \end{array}\right.$$

It was obtained by consistency inspection of all secondary indicators:$$\left\{\begin{array}{l} \beta 1={\left(CI,\;CR\right)}^{T}={\left(\begin{array}{cc} 0.001 & 0.002 \end{array}\right)}^{T}\\ \beta 2={\left(CI,\;CR\right)}^{T}={\left(\begin{array}{cc} 0.003 & 0.005 \end{array}\right)}^{T}\\ \beta 3={\left(CI,\;CR\right)}^{T}=\;{|\left(\begin{array}{cc} 0 & 0 \end{array}\right)}^{T}\\ \beta 4={\left(CI,\;CR\right)}^{T}={\left(\begin{array}{cc} 0.001 & 0.002 \end{array}\right)}^{T} \end{array}\right.$$

According to the consistency inspection principle, it was required that CR<0.1 to meet the consistency. It can be seen that the consistency ratio CR of the secondary indicators was less than 0.1, indicating that the judgment matrices B1, B2, B3 and B4 had satisfactory consistency. After inspection, the weight of secondary indicators is shown in Figure 10.

By formula $S_{i}=\beta_{i}\times B_{k}\left(i=1,2,3;k=1,2,3,4,5\right)$ the overall normalized weight vector of each secondary indicator based on the primary indicator was calculated according to the above analysis, the integrated weight was obtained, as shown in Table 9.

In Figure 11, it is easy to find that “ventilation conditions”, “secondary hazards”, “preparedness” and “geographical conditions” accounted for a larger proportion of the area in the distribution than other factors, indicating that these four indicators had a greater impact on the fire risk of lithium batteries.

### 4.4. Risk Evaluation

A factor set representing multiple factors of comprehensive evaluation was established according to the influencing factors obtained previously: (15)$$U_{i}=\left\{B_{11}B_{12}B_{13}B_{21}B_{22}B_{23}B_{31}B_{32}B_{33}B_{34}B_{41}B_{42}B_{43}\right\}$$

Then the evaluation set composed of multiple decisions was established:(16)$$V=\left\{V_{1}V_{2}V_{3}V_{4}V_{5}\right\}$$

Wherein, *V* = {v_1_ v_2_ v_3_ v_4_ v_5_} represents the evaluation criteria that affect the fire hazard of lithium battery, respectively, “particularly important”, “very important”, “relatively important”, “generally important” and “slightly important”, corresponded to the “special danger”, “highly dangerous”, “significant danger”, “general danger” and “low danger” of the fire hazard of lithium battery.

Because each factor in the factor set had different effects on the evaluation, combining the weights (non-combination weights) of the decision-making layer obtained above, we obtained:(17)$$Q=\left\{\beta_{1}^{T}\beta_{2}^{T}\beta_{3}^{T}\beta_{4}^{T}\right\}$$

Moreover, the comments were not absolute affirmation and negation, so the comprehensive evaluation could be considered as a fuzzy set on, which was recorded as: (18)$$M=\left(C_{1}C_{2}C_{3}C_{4}\right)\in F(V)$$

In the formula, *C_m_* shows the position of the m-th comment in the overall evaluation V.

B1, B2, B3 and B4 represent different subsets of indicators in the three-level comprehensive evaluation indicator system in Figure 3. Each Bi (i = 1,2,3,4) was subjected to fuzzy evaluation, respectively. The index B*_ij_* under Bi (i = 1,2,3,4) was considered separately. The degree of the kth comment *v_k_* belonging to k was obtained by the Delphi method, and a fuzzy evaluation matrix R under Bi (i = 1,2,3,4) was obtained. See Table 10 for details.

The evaluation data came from the actual survey, that is, each indicator was evaluated according to the five levels of the evaluation criteria, and the interviewees made choices according to their own experience and feelings, so as to obtain the evaluation matrix. A total of 100 evaluation questionnaires were sent out within the scope of the university, 78 were returned and 70 were valid. It should be noted that this survey was different from the questionnaire on the fire risk of lithium battery in the mine. It was the third party’s evaluation of the fire risk of lithium battery in the mine. Each indicator was divided into five grades according to the evaluation set, that is, five options. The interviewees chose one of them according to their own feelings.

The fuzzy set on V was calculated by using the ratio of the fuzzy evaluation matrix and the decision-making layer weight (non-combination weight) obtained above. According to: 


$$\left(\begin{array}{c} W_{i}\cdot R_{i}=\left(\omega_{1},\omega_{2},\omega_{3},\ldots ,\omega_{m}\right)\left(\begin{array}{cccc} r_{11} & r_{12} & \ldots & r_{1m}\\ r_{21} & r_{22} & \ldots & r_{2m}\\ \vdots & \vdots & \ddots & \vdots \\ r_{n1} & r_{n2} & \ldots & r_{nm} \end{array}\right)\\ \beta 1={\left(B11\;B12\;B13\right)}^{T}={\left(\begin{array}{ccc} 0.246 & 0.477 & 0.277 \end{array}\right)}^{T}\\ \beta 2={\left(B21\;B22\;B23\right)}^{T}={\left(\begin{array}{ccc} 0.150 & 0.603 & 0.247 \end{array}\right)}^{T}\\ \beta 3=\left(B31\;B32\;B33\;B34\right)={\left(\begin{array}{cccc} 0.371 & 0.213 & 0.180 & 0.236 \end{array}\right)}^{T}\\ \beta 4={\left(B41\;B42\;B43\right)}^{T}={\left(\begin{array}{ccc} 0.544 & 0.265 & 0.191 \end{array}\right)}^{T} \end{array}\right.$$


The evaluation matrix of the evaluation set *V* = {v_1_ v_2_ v_3_ v_4_} formed by decision with formula $\omega =\beta^{T}$ was obtained.



$C_{i}=\left[\begin{array}{l} 0.2516\;0.3212\;0.2702\;0.1435\;0.0218\\ 0.2572\;0.2130\;0.2142\;0.1387\;0.1763\\ 0.2718\;0.2747\;0.2320\;0.1278\;0.0936\\ 0.1994\;0.1924\;0.3255\;0.2017\;0.0809 \end{array}\right]$



For the set $U=\left\{B_{1},B_{2},B_{3},B_{4}\right\}$ comprehensive evaluation, the known weight vector was *W* = (0.144, 0.196, 0.258, 0402), and the comprehensive evaluation matrix was obtained.
C = (0.237, 236, 0.272, 0.162, 0.094)

According to the evaluation criteria in {v_1_ v_2_ v_3_ v_4_ v_5_}, “particularly important”, “very important”, “relatively important”, “generally important” and “slightly important” corresponded to the “special danger”, “high danger”, “significant danger”, “general danger” and “low danger” of lithium battery fire hazard, respectively. According to the specified quantitative values of each element in the evaluation set, *v*_1_ = 100, *v*_2_ = 85, *v*_3_ = 70, *v*_4_ = 55, *v*_5_ = 40. Under this mathematical evaluation model, the closer the fire risk score of lithium battery was to 100, the more dangerous it was and the closer to 40, the safer.

Therefore, under the current assessment conditions, the fire risk of lithium battery in the mine could be calculated as:


*δ* = *c*1 × *v*1 + *c*2 × *v*2 + *c*3 × *v*3 + *c*4 × *v*4 + *c*5 × *v*5 
= 0.237 × 100 + 0.236 × 85+0.272 × 70+0.162 × 55+0.094 × 40 
= 75.37


According to fire safety inspection of the daily operation of the mine, there were obvious problems in firefighting facilities, safety management system, instrumentation operation and special equipment, indicating that the possibility of lithium battery fire was still high in practice. Our assessment of the lithium battery fire in the mine using the optimized FAHP method was 75.37, which meant that the mine’s lithium battery fire risk effect was at the second level of “high risk” and had a high match with the actual situation in the mine.

### 4.5. Fire risk Response

#### 4.5.1. Fire Preventing Measures

(1) Regularly carry out safety training for staff to improve their comprehensive safety quality. Standardize the operation of underground drivers to avoid accidents caused by improper use. Carry out a series of activities of emergency rescue and disposal ability competition to effectively improve personal emergency ability and facilitate escape in case of accidents.

(2) Find a balance between economic benefits and safety performance and select battery products with less risk after accidents. The lithium battery in-situ monitoring system is set up to detect dangerous signals in time and nip the crisis in the cradle.

(3) Improve the reducing force, enhance the preparation, improve the responsiveness, and protect the resilience. By arranging reasonable fire-fighting facilities, the impact of early warning and alarm devices can be avoided and reduced to the greatest extent. Control the crisis by formulating crisis disposal plan, establishing crisis early warning mechanism, building crisis early warning system, cultivating crisis management team, conducting crisis training and drills, and help decision-makers make rapid decisions through intuitive information and evaluation. Effectively deal with the crisis by responding and making decisions in a short time, comprehensively obtain first-hand information from multiple parties, provide the basis for disposal, demonstrate the loss reduction plan, and improve the optimal solution. Through risk assessment, analyze the harmfulness to individuals and institutions, quantitatively assess the time and cost required to recover the harmful effects of the crisis, formulate effective measures, and eliminate the adverse effects in the shortest time.

(4) Under the condition of controlling production cost, increase the wind speed in the roadway as much as possible. Emergency escape routes and refuge chambers shall be arranged scientifically and reasonably. Establish fire area isolation zone to prevent secondary hazards caused by fire.

#### 4.5.2. Risk Management Implementation

The lithium fire accidents in coal mines are the result of the interaction and incongruity among human, machine, environment and management. The adopted response measures were to use the idea of risk early warning to promote the systematization and standardization of measures by systematization, so as to avoid the risks that affect the safe production of coal mines. The specific risk warning process is shown in Figure 12. In the process of the whole risk early warning process, the FED model was constantly used to calculate the optimal interval of the characterization factors, and then applied to the FAHP model to continuously optimize the emergency measures, so as to finally achieve the highest effect of the measures, thereby improving the ability of the coal mine to avoid risks, so that the coal mine could achieve the goal of safe production.

#### 4.5.3. Risk Factor Control

According to the above analysis results, ventilation conditions contributed the most to the risk of lithium battery, accounting for 21.9%. Reasonable regulation of tunnel wind speed played a vital role in reducing the risk of lithium battery fire.

According to the acceptable living environment regulations issued by NFPA, the average volume fraction of CO when a person was in a fire environment for more than 15min did not exceed 0.05%. In this paper, the critical volume fraction of CO hazard was selected as 0.05%, and it was considered that the time from the time of fire to the time when the smoke with CO volume fraction of 0.05% chased the escape personnel was the available safe evacuation time (ASET). FDS software was used for numerical simulation to build the relationship between wind speed and CO migration velocity. When 1.0 m/s < *v*_air_ < 2.4 m/s, the CO migration velocity *v*_co_ was linear with the wind velocity *v*_air_ as shown in Figure 13, and the fitting equation was:(19)$$v_{\mathrm{co}}=-0.06+1.09v_{\mathrm{air}}$$

According to personnel escape speed and CO migration speed, the available evacuation time was expressed as:(20)$$v_{\mathrm{move}}\left(t_{\min}-t_{\det}-t_{pre}\right)-L_{\mathrm{face}}=L_{500}+v_{\mathrm{co}}\left(t_{\min}-t_{500}\right)$$

Where, *t_min_* is the available safe evacuation time, s; *v*_move_ is the escape speed of personnel, *v*_move_ = 1 m/s; *v*_co_ is CO migration velocity, m/s; *t*_det_ is the fire detection time, *t*_det_ = 30 s; *t_pre_* is the preparation time for evacuation *t_pre_* = 30 s; *L_face_* is the length of mining face, *L_face_* = 150m; *L*_500_ is the distance from the mining face at 300 m at the downwind side of the fire source, *L*_500_ = 150 m; *t*_500_ is the time taken for CO volume fraction to reach 0.05% at 300m from the downwind side of the fire source and 1.5 m from the bottom plate.

After calculation, the available safe evacuation time *t*_min_ is shown in Figure 14. The higher the wind speed, the smaller the *t*_min_: if the wind speed was very close to the escape speed, the evacuation risk was low, and when the wind speed was high, the evacuation time was tight; with the increase of wind speed, its influence on *t*_min_ became weaker. When the critical wind speed was 2.0 m/s, the limit available time was the shortest and the fire risk was the highest. Therefore, the wind speed of the roadway was close to the escape speed of the personnel in the fire period of 1m/s, and had a certain difference from the critical wind speed of 2 m/s. Combined with the effectiveness of ventilation, sewage discharge, temperature reduction and dust removal in the roadway, the roadway wind speed of 1.2 m/s effectively reduced the fire risk of lithium battery on the premise so that other functions were not greatly affected.

## 5. Conclusions

In order to reveal the potential coupling hazards of thermal runaway injury and toxic smoke poisoning caused by large capacity lithium battery for mining, a multiple risk assessment model for the possible occurrence of lithium battery fire disaster in the confined space of the mine was constructed in this paper. In order to reduce the subjective deviation caused by the commonly used expert scoring method to determine the weight of the factors at the decision-making level, this paper proposed an innovative analysis method that combined the quantitative factor characterization of hazard sources and fuzzy analytic hierarchy process.

1. A quantitative characterization model of multiple hazard sources, namely, lithium battery thermal runaway and poison gas diffusion in the tunnel, was proposed. Experiments and numerical methods were implemented to reveal the mechanism and risk of heat and mass disaster propagation under catastrophic conditions. Based on the FED asphyxiation model, the contribution values of risk factors from decision-making levels were calculated quantitatively.

2. Based on the quantitative factor characterization model, the FAHP was introduced to apply to the quantitative assessment of the potential risk of lithium battery fire in the specific mine working face, and the risk levels of 13 hazard sources in four categories were identified. The construction matrix was used to verify the scientificity and rationality of the quantitative assignment method. Taking a typical mine in Shaanxi Province in China as a study case, the result is classified as high risk mine for lithium battery fire, and the early warning model and risk management implementation were discussed.

3. Aiming at the risk factor that was most likely to to occur and had the highest probability of 21.9%, namely, the ventilation condition (smoke diffusion) in the mine environmental factors, the monitoring threshold of ventilation parameters under the control of safe evacuation time was established through FDS simulation, and the optimal air supply model in the working face was proposed from the perspective of mine lithium battery fire risk prevention and control.

## Figures and Tables

**Figure 1 ijerph-20-00456-f001:**
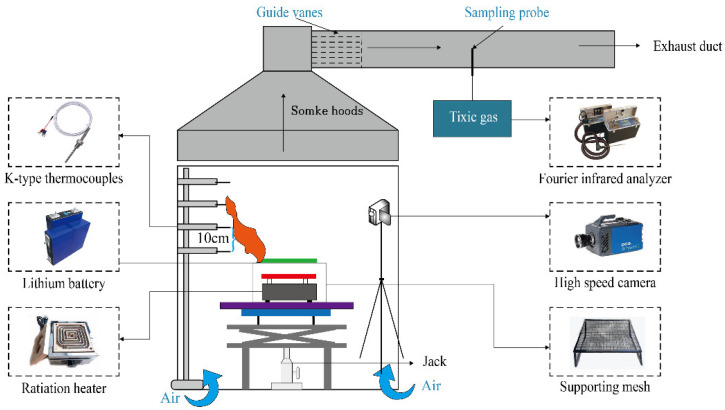
Experiment setup.

**Figure 2 ijerph-20-00456-f002:**
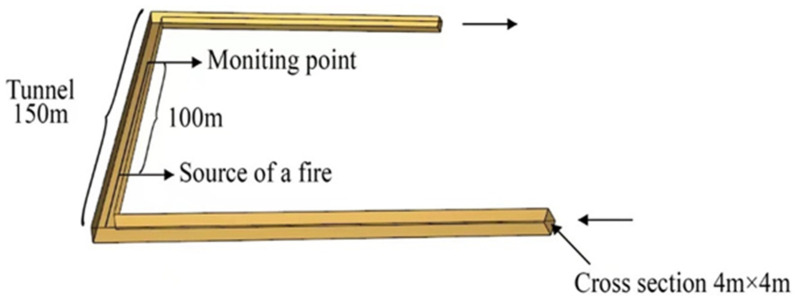
Physical model of tunnel fire.

**Figure 3 ijerph-20-00456-f003:**
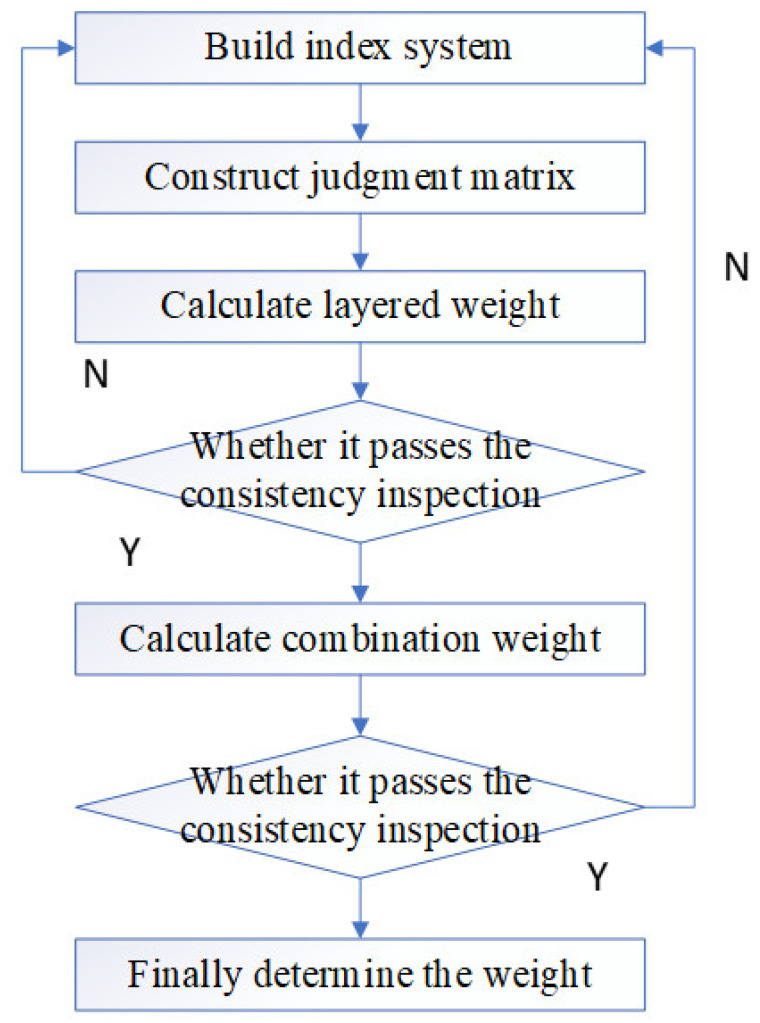
Flow chart of fuzzy analytic hierarchy process.

**Figure 4 ijerph-20-00456-f004:**
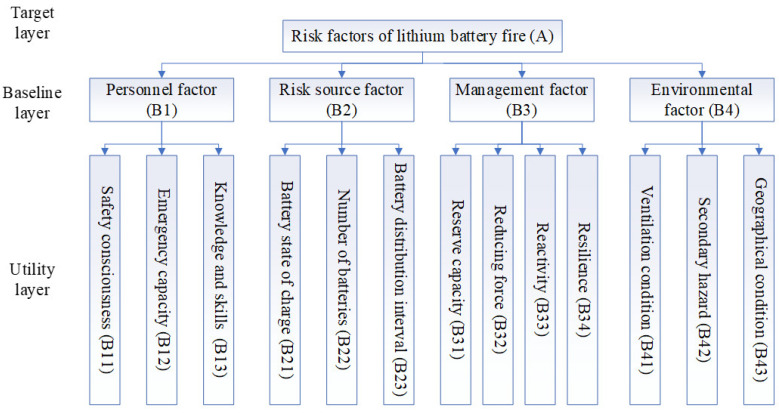
Structural and hierarchical topological relationship of lithium battery fire risk factors.

**Figure 5 ijerph-20-00456-f005:**
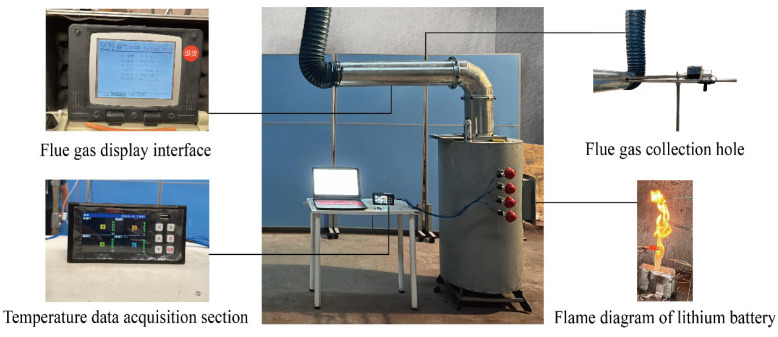
Experimental details display diagram.

**Figure 6 ijerph-20-00456-f006:**
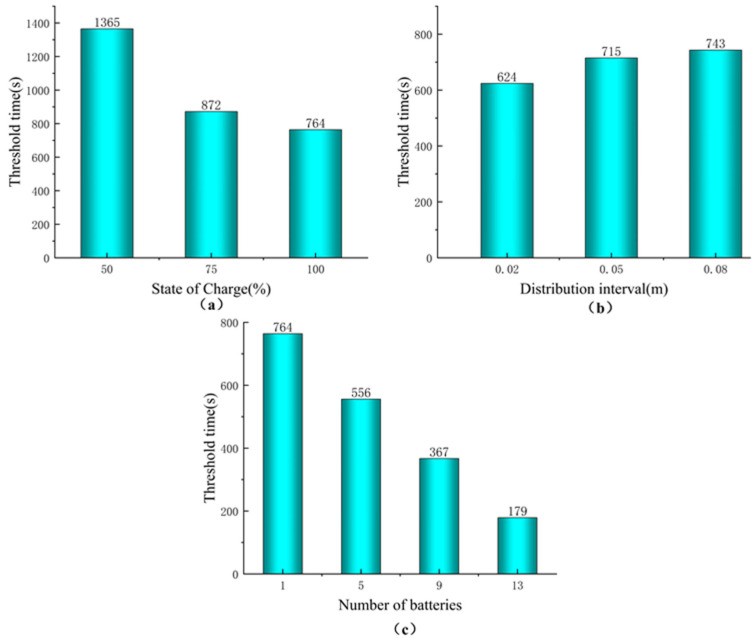
Corresponding diagram of hazard factor and time to threshold. (**a**) State of Charge vs. Threshold Time; (**b**) Distribution interval and threshold time diagram; (**c**) Number of batteries versus threshold time.

**Figure 7 ijerph-20-00456-f007:**
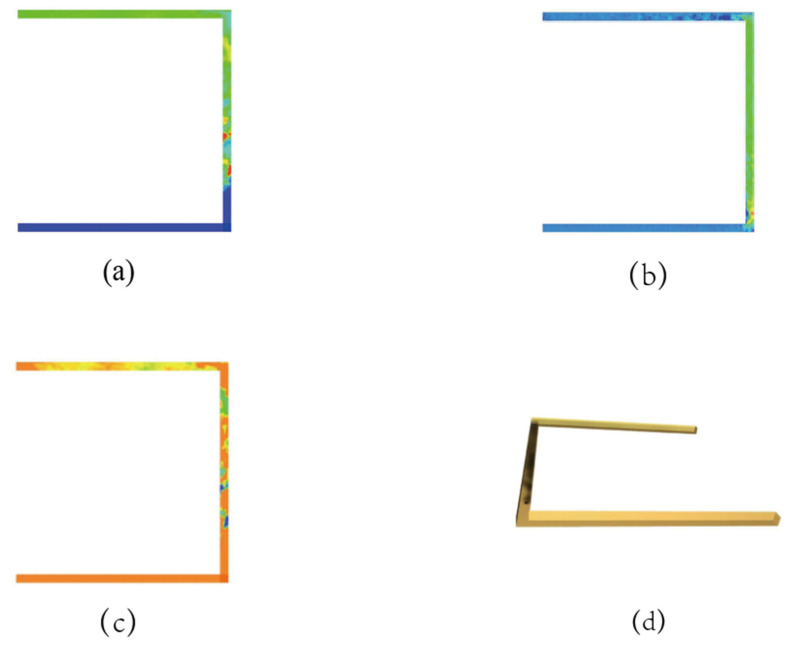
Cloud chart of simulation results. (**a**) CO concentration nephogram; (**b**) Velocity nephogram; (**c**) Visibility nephogram; (**d**) Smoke space nephogram.

**Figure 8 ijerph-20-00456-f008:**
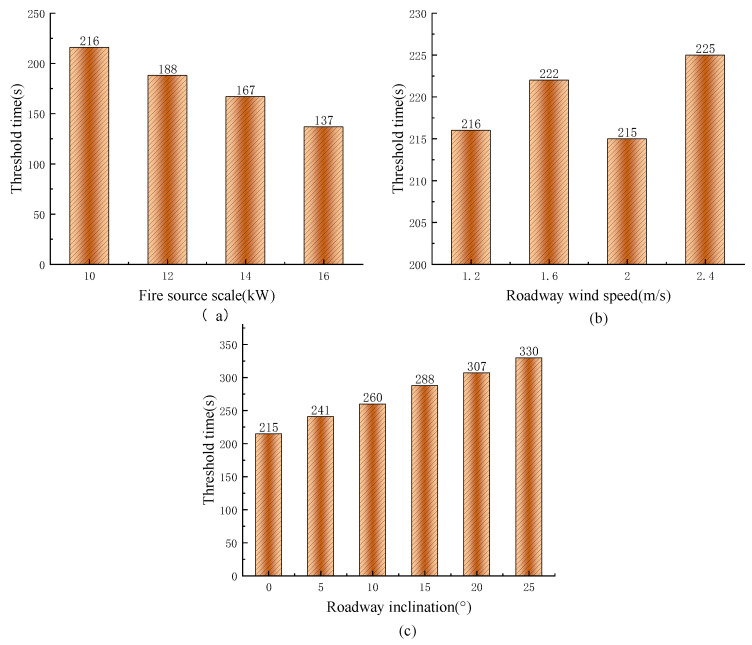
Corresponding diagram of various factors of place factors and time to reach threshold. (**a**) Diagram of fire source scale and threshold time; (**b**) Diagram of tunnel wind speed and threshold time; (**c**) Diagram of roadway dip angle and threshold time.

**Figure 9 ijerph-20-00456-f009:**
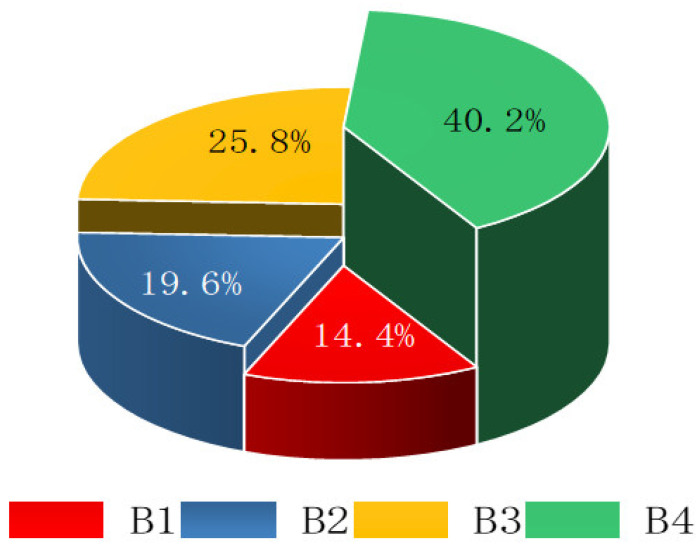
Index chart of weighting coefficient of primary indicators.

**Figure 10 ijerph-20-00456-f010:**
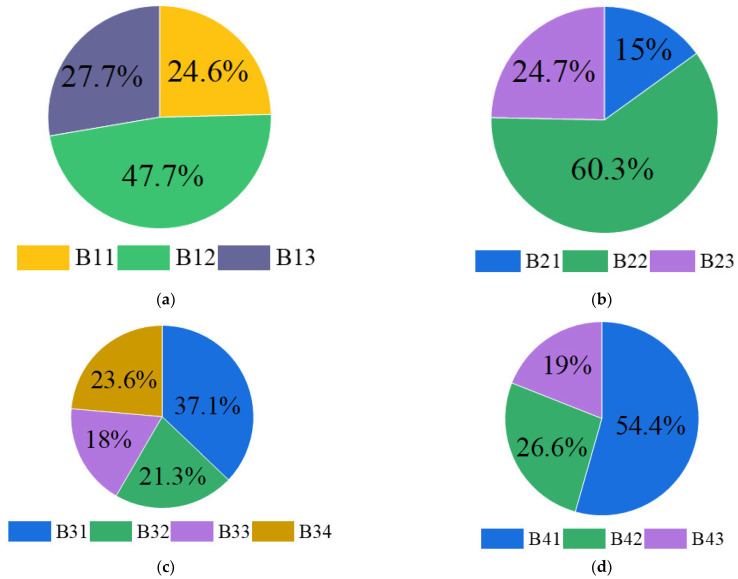
Weights of secondary indicators. (**a**) Personnel factor; (**b**) Risk source factor; (**c**) Crisis management factor; (**d**) Environmental factor.

**Figure 11 ijerph-20-00456-f011:**
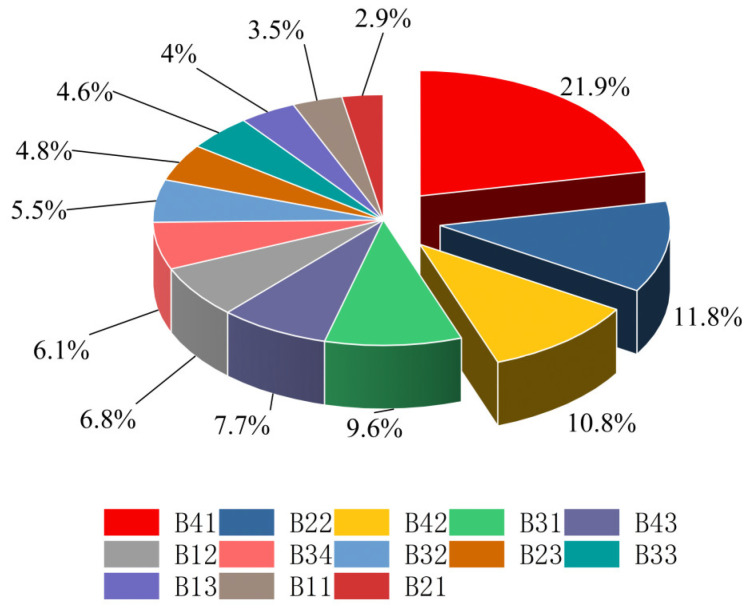
Integrated weight index.

**Figure 12 ijerph-20-00456-f012:**
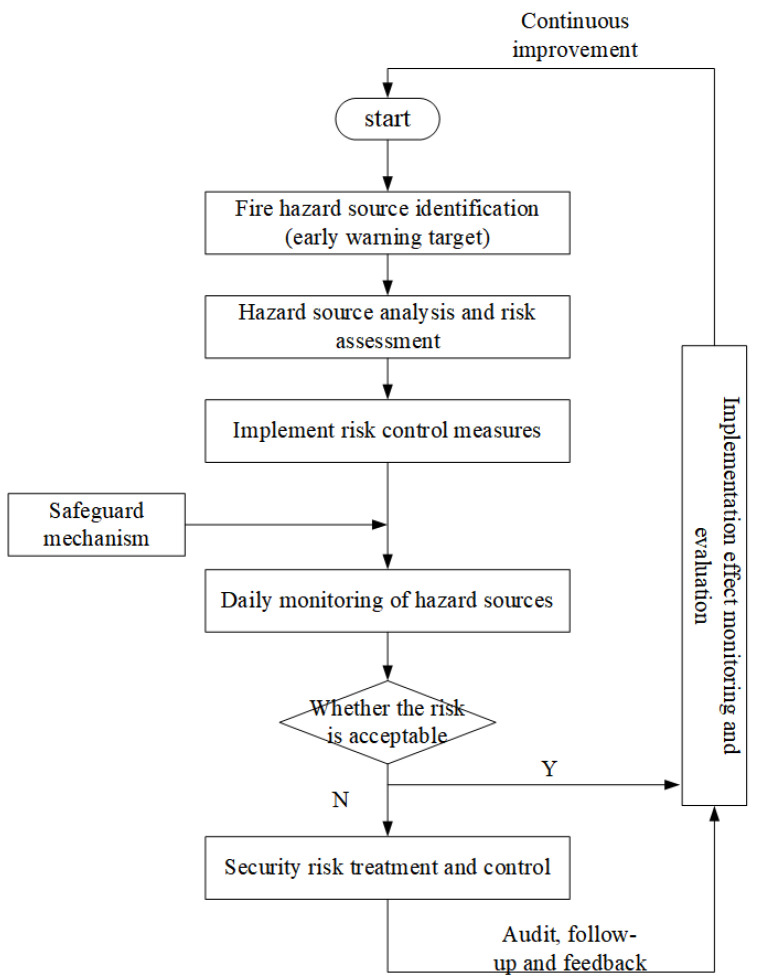
Risk response implementation flow chart.

**Figure 13 ijerph-20-00456-f013:**
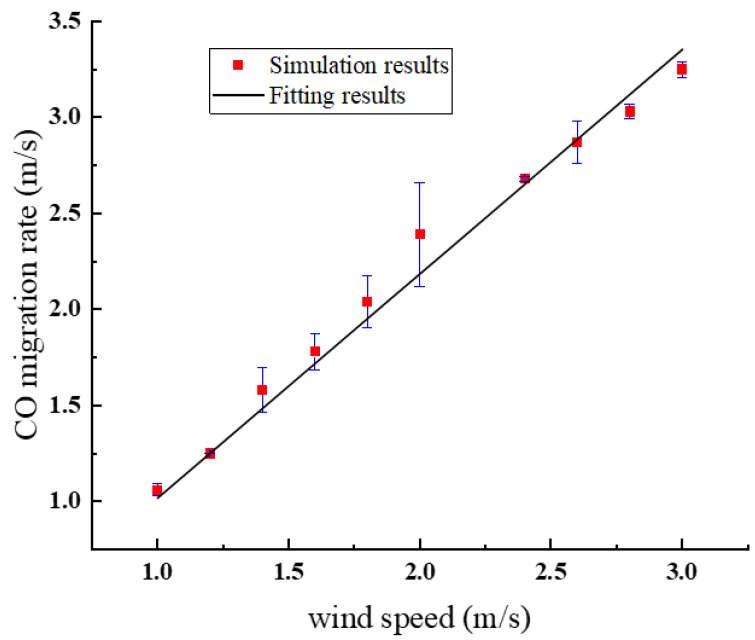
Relationship between CO migration speed and wind speed.

**Figure 14 ijerph-20-00456-f014:**
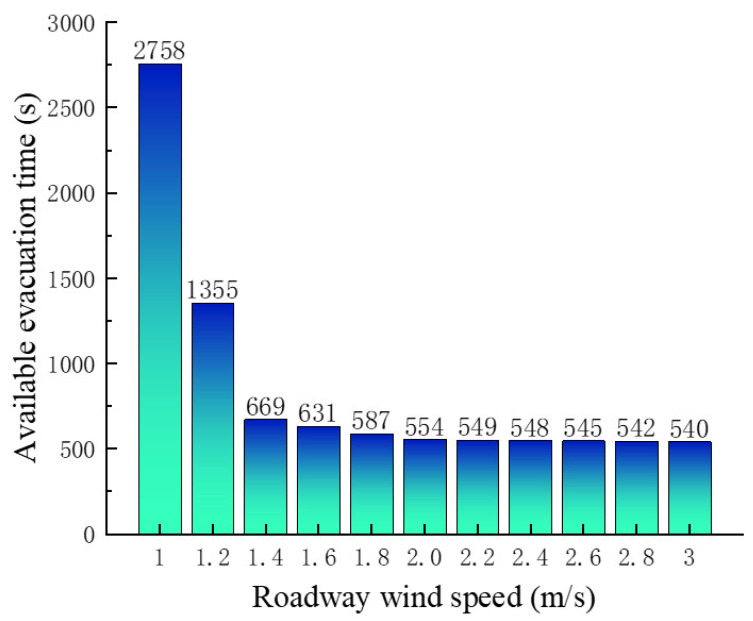
Trend chart of available evacuation time.

**Table 1 ijerph-20-00456-t001:** Comparison of advantages and disadvantages of risk assessment methods.

Risk Assessment Method	Advantages and Disadvantages	
LEC evaluation method (LEC stands for likelihood, risk exposure frequency, consequences)	Advantages	1. Simple and easy
		2. Semi quantitative evaluation method
	Disadvantages	1. The scores of L, E and C can only represent a certain range
		2. The score judgment is related to the work experience of the evaluators
MES evaluation method (MES stands for measures, exposure status, severity of consequences)	Advantages	1. Information collection is easy to obtain
		2. High matching degree of construction project pre-assessment
		3. Qualitative classification
	Disadvantages	1. High professional requirements
		2. Limitation of application scope
Index evaluation method	Advantages	1. Easy to use
		2. It is applicable to the fields where multiple disasters coexist, the structure is complex, and the probability of hidden danger is difficult to determine
	Disadvantages	1. Neglect the function of system security assurance system
		2. Ignore the importance difference between factors
		3. Poor flexibility and sensitivity
		4. The scope of additional systems for risk factors is too wide and there are too many compensation items
Fuzzy comprehensive evaluation method	Advantages	Be good at multi factor index synthesis and multi-index problems
	Disadvantages	1. The determination and change of factor weights need to be addressed
		2. The distribution of weights almost depends on manual methods
Probabilistic risk assessment	Advantages	1. Perfection of theoretical basis
		2. Strong portability
		3. Objective and accurate description
		4. Quantitative evaluation method
	Disadvantages	1. Many uncertain factors
		2. Unclear factor status
		3. Difficulty in coefficient complete decomposition

**Table 2 ijerph-20-00456-t002:** Comparison scale.

Judgment Scale	Significance
1	Factor $a_{i}$ is as important as factor $a_{j}$
2	Importance lies between the upper and lower scales
3	Factor $a_{i}$ is more important than factor $a_{j}$
4	Importance lies between the upper and lower scales
5	Factor $a_{i}$ is very important than factor $a_{j}$

**Table 3 ijerph-20-00456-t003:** RI value of random consistency index.

Order n	1	2	3	4	5	6	7	8	9
RI value	0	0	0.52	0.89	1.12	1.26	1.36	1.41	1.46

**Table 4 ijerph-20-00456-t004:** Fire risk assessment indexes of lithium battery.

Level I Indicators	Level II Indicators
Personnel factor (B1)	Safety consciousness (B11)
	Emergency capacity (B12)
	Knowledge and skill quality (B13)
Risk source factor (B2)	State of Charge (B21)
	Number of lithium batteries (B22)
	Distribution interval (B23)
Crisis management factor (B3)	Reserve capacity (B31)
	Reducing force (B32)
	Reactivity (B33)
	Resilience (B34)
Environmental factor (B4)	Ventilation conditions (B41)
	Secondary hazard (B42)
	Geographical conditions (B43)

**Table 5 ijerph-20-00456-t005:** Comparison matrix of primary indicators.

Average Value	Project	B1	B2	B3	B4
1.400	B1	1	0.737	0.560	0.359
1.900	B2	1.357	1	0.760	0.487
2.500	B3	1.786	1.358	1	0.641
3.900	B4	2.786	2.053	1.560	1

**Table 6 ijerph-20-00456-t006:** Corresponding table of level I indicators.

Project	Weight	Maximum Characteristic Value	CI Value
Personnel factors B1	0.144	4.008	0.003
Risk source factors B2	0.196		
Crisis management factors B3	0.258		
Environmental factors B4	0.402		

**Table 7 ijerph-20-00456-t007:** Contribution rate and weight distribution of hazard indicators.

Secondary Indicators	Contribution Rate	Weight Distribution
B21	12.02	0.150
B22	48.73	0.603
B23	19.83	0.247
B41	12.33	0.544
B42	6.43	0.266
B43	4.60	0.190

**Table 8 ijerph-20-00456-t008:** Judgment matrix of secondary indicators.

$\omega_{B_{1}}=\left(\begin{array}{ccc} 1 & 0.516 & 0.889\\ 1.938 & 1 & 1.722\\ 1.125 & 0.581 & 1 \end{array}\right)$	$\omega_{B_{2}}=\left(\begin{array}{ccc} 1 & 0.25 & 0.61\\ 4 & 1 & 2.46\\ 1.65 & 0.41 & 1 \end{array}\right)$
$\omega_{B_{3}}=\left(\begin{array}{cccc} 1 & 1.737 & 2.063 & 1.571\\ 0.576 & 1 & 1.188 & 0.905\\ 0.485 & 0.842 & 1 & 0.762\\ 0.636 & 1.105 & 1.313 & 1 \end{array}\right)$	$\omega_{B_{4}}=\left(\begin{array}{ccc} 1 & 2.05 & 2.86\\ 0.49 & 1 & 1.39\\ 0.35 & 0.72 & 1 \end{array}\right)$

**Table 9 ijerph-20-00456-t009:** Consolidated Weight Table.

Target Layer (A)	Criterion Layer (B)	Weight	Decision Makers(P)	Weight	Combination Weight
A	B1	14.4%	B11	24.6%	3.5%
			B12	47.7%	6.8%
			B13	27.7%	4%
	B2	19.6%	B21	15%	2.9%
			B22	60.3%	11.8%
			B23	24.7%	4.8%
	B3	25.8%	B31	37.1%	9.6%
			B32	21.3%	5.5%
			B33	18%	4.6%
			B34	23.6%	6.1%
	B4	40.2%	B41	54.4%	21.9%
			B42	26.6%	10.8%
			B43	19%	7.7%
SUM					1

**Table 10 ijerph-20-00456-t010:** Evaluation matrix.

$R_{1}=\left[\begin{array}{ccccc} 0.243 & 0.2 & 0.357 & 0.171 & 0.029\\ 0.286 & 0.371 & 0.2 & 0.129 & 0.014\\ 0.2 & 0.343 & 0.314 & 0.144 & 0.029 \end{array}\right]$ $R_{3}=\left[\begin{array}{ccccc} 0.143 & 0.2 & 0.314 & 0.186 & 0.157\\ 0.286 & 0.371 & 0.2 & 0.129 & 0.014\\ 0.371 & 0.3 & 0.143 & 0.043 & 0.143\\ 0.386 & 0.286 & 0.2 & 0.1 & 0.028 \end{array}\right]$	$R_{2}=\left[\begin{array}{ccccc} 0.171 & 0.114 & 0.186 & 0.2 & 0.329\\ 0.343 & 0.243 & 0.186 & 0.157 & 0.07\\ 0.1 & 0.2 & 0.3 & 0.057 & 0.343 \end{array}\right]$ $R_{4}=\left[\begin{array}{ccccc} 0.2 & 0.2 & 0.3 & 0.243 & 0.057\\ 0.157 & 0.1 & 0.386 & 0.2 & 0.157\\ 0.257 & 0.3 & 0.314 & 0.086 & 0.043 \end{array}\right]$

## Data Availability

Data are readily available at request.

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
