# Peer review of "Multi-Risk Assessment of Mine Lithium Battery Fire Based on Quantitative Factor Characterization"

_ijerph, 2022, doi:10.3390/ijerph20010456_

Round 1
Reviewer 1 Report
This is an interesting work, well structured and detailed.
The introduction is good and justifies the significance of the work.
Minor mistakes that need to be checked.
I would expect a more extensive literature review describing what has been done so far in the field of lithium battery fire risk assessment, as well as how and where FAHP has been used to conduct risk assessment analyses.
Some remarks on how to better describe and illustrate the methodology and its implementation are made in the annotated copy of the manuscript.
It is suggested that the conclusions section is updated, to focus more on the significance of this work, how it can be implemented immediately, and that other mining industry stakeholders (except for coal producers) can use this methodology.

Author Response
Point 1: I would expect a more extensive literature review describing what has been done so far in the field of lithium battery fire risk assessment, as well as how and where FAHP has been used to conduct risk assessment analyses.
Response 1: In the introduction, we are reviewed the risk assessment of lithium batteries. The FAHP method and its scope of application are elaborated in 2.2.
Point 2: Some remarks on how to better describe and illustrate the methodology and its implementation are made in the annotated copy of the manuscript.
Response 2: We have modified the content you marked in the article and marked the modified part in red in the original text.
- We corrected the spelling mistakes in the paper and corrected the language of the paper again.
- We have revised the parts that may cause misunderstanding.
- Line 142, the source of 35000 is supplemented.
- Revise the abbreviations you mentioned one by one
- Line 343, you asked about the standard division of different levels of AHP. We supplemented the risk identification method in 2.1. The source of the standard is mainly combined with the existing literature analysis method and expert evaluation method to divide the primary indicators of lithium battery from four aspects of man-machine management ring, and then the secondary indicators are divided by using the correlation degree and mutual independence criteria. With regard to the accuracy of the three secondary indicators, we screened many indicators based on literature analysis, and then used questionnaires and expert evaluation methods to screen out highly relevant indicators with good accuracy. '' With reference to the risk identification methods in 2.1 and the literature on the study of exogenous fires in coal mines using the FAHP method, and taking into account the characteristics of lithium battery fires, we use the literature analysis method to identify the risk indicators for lithium battery fires in mines, and the identified risk factors are classified and then the opinions and suggestions of relevant experts and scholars are sought in the form of interviews to screen out the unreasonable factors. The primary indicators are classified from the four aspects of human-machine environmental management for the risk of lithium battery fires in coal mines, and the secondary indicators are obtained by screening according to the correlation and mutual independence criteria with the primary indicators. For the case mine, we have analyzed various risk factors for lithium battery fires in the mine based on the power feedback from workers, underground vehicle conditions, mine heat damage assessment, and roadway layout and management, as shown in Table 4.’’
- Line 399, the picture is optimized
- Line 413, we have taken your suggestions to supplement and optimize the table
Point 3: It is suggested that the conclusions section is updated, to focus more on the significance of this work, how it can be implemented immediately, and that other mining industry stakeholders (except for coal producers) can use this methodology.
Response 3: We have updated the conclusion and added the method implementation flow chart 12 to facilitate the implementation of other relevant practitioners.
Special thanks to you for your good comments.

Reviewer 2 Report
This is a good approach to a novel research problem.
Author Response
Thank you for your review and comments.

Reviewer 3 Report
This work present assessment of the durability of lithium batteries in the mine and mining facilities, as well as of the phase in which they become risky and the and associated influencing factors. The work is well structured and the research is compatible with the requirements for risk research in the relevant field.
In the introduction, the literature review of the current status of the lithium battery leakage risk problem and the corresponding consequences is insufficient.
Please add a bit more safety conclusions in the last part of the manuscript and exactly what this work has contributed to.
In order to complete this work, it is good to say a few words about the reliability of the Risk assessment model and models that are used.
Author Response
Point 1: In the introduction, the literature review of the current status of the lithium battery leakage risk problem and the corresponding consequences is insufficient.
Response 1: In the introduction, we supplemented the literature review on the current situation of lithium battery leakage risk and the insufficient corresponding consequences.
Point 2: Please add a bit more safety conclusions in the last part of the manuscript and exactly what this work has contributed to.
Response 2: We have added "4.5. Risk Response" to supplement the conclusion and added " Taking a typical mine in Shaanxi Province in China as a study case, the result is classi-fied as high risk mine for lithium battery fire, and the early warning model and risk management implementation are discussed." in the conclusion.
Point 3: In order to complete this work, it is good to say a few words about the reliability of the Risk assessment model and models that are used.
Response 3: " According to fire safety inspection of the daily operation of the mine, there are obvious problems in firefighting facilities, safety management system, instrumentation operation, and special equipment, indicating that the possibility of lithium battery fire is still high in practice. Our assessment of the lithium battery fire in the mine using the optimized FAHP method was 75.37, which means that the mine's lithium battery fire risk effect is at the second level of "high risk" ,and has a high match with the actual situation in the mine. "since it was added, it indicates the reliability of the model.
Special thanks to you for your good comments.

Round 2
Reviewer 1 Report
I am pleased with the changes the authors made.
The manuscript has been improved significantly and is now worth being published.